# An Integrated Structural Air Electrode Based on Parallel Porous Nitrogen-Doped Carbon Nanotube Arrays for Rechargeable Li–Air Batteries

**DOI:** 10.3390/nano9101412

**Published:** 2019-10-03

**Authors:** Yu Li, Zhonglin Zhang, Donghong Duan, Yunxia Han, Kunlei Wang, Xiaogang Hao, Junwen Wang, Shibin Liu, Fanhua Wu

**Affiliations:** College of Chemistry and Chemical Engineering, Taiyuan University of Technology, Taiyuan 030024, China; liyu@tyut.edu.cn (Y.L.); zlzhang@tyut.edu.cn (Z.Z.); duandonghong@tyut.edu.cn (D.D.); 15513019359@163.com (K.W.); xghao@tyut.edu.cn (X.H.); wangjunwen@tyut.edu.cn (J.W.); fhwu_0519@nuc.edu.cn (F.W.)

**Keywords:** nitrogen-doped carbon nanotube array, air electrode, electrode overpotential, Li–air battery, discharge–charge performance

## Abstract

The poor discharge and charge capacities, and the cycle performance of current Li–air batteries represent critical obstacles to their practical application. The fabrication of an integrated structural air electrode with stable parallel micropore channels and excellent electrocatalytic activity is an efficient strategy for solving these problems. Herein, a novel approach involving the synthesis of nitrogen-doped carbon nanotube (N-CNT) arrays on a carbon paper substrate with a conductive carbon-black layer for use as the air electrode is presented. This design achieves faster oxygen, lithium ion, and electron transfer, which allows higher oxygen reduction/evolution reaction activities. As a result, the N-CNT arrays (N/C = 1:20) deliver distinctly higher discharge and charge capacities, 2203 and 186 mAh g^−1^, than those of active carbons with capacities of 497 and 71 mAh g^−1^ at 0.05 mA cm^−2^, respectively. A theoretical analysis of the experimental results shows that the parallel micropore channels of the air electrode decrease oxygen diffusion resistance and lithium ion transfer resistance, enhancing the discharge and charge capacities and cycle performance of Li–air batteries. Additionally, the N-CNT arrays with a high pyridinic nitrogen content can decompose the lithium peroxide product and recover the electrode morphology, thereby further improving the discharge–charge performance of Li–air batteries.

## 1. Introduction

Metal–air batteries, such as Li–air batteries, have high theoretical energy and power densities, making them ideally suitable for novel electric vehicle and portable power applications [1,2,3]. However, so far, metal–air batteries have not been able to fulfill the practical energy stability, power stability, and safety requirements of energy conversion and storage systems. The discharge and charge capacities, and the cycle performance of Li–air batteries remain low in current operations, primarily because of the complex problems of the Li anode, electrolyte, membrane, and air electrode [4,5]. Over the past ten years, research interest in Li anode, electrolyte, and membrane materials, all of which have considerable limitations in realizing practical applications, has increased notably [6,7,8,9]. More importantly, long-standing previous research has shown that the air electrode is the key component in terms of improving the performance of a Li–air battery, mainly because the oxygen electro-reduction/electro-oxidation processes occur at this component. Oxygen, lithium ions, and electrons are also simultaneously transferred in the air electrode, and resistance to the transfer of these components can severely hinder the oxygen reduction/evolution reaction (ORR/OER) kinetics [10,11,12]. In addition, any residual solid lithium peroxide (Li_2_O_2_) product that accumulates in the pores of the air electrode observably restricts the intrinsic kinetics of the oxygen electro-reduction/electro-oxidation processes and the transfer dynamics of oxygen, and greatly increases the transfer resistance of lithium ions and electrons [13]. Thus, improving the air electrode remains a significant challenge in the commercialization of Li–air batteries [14].

To date, numerous studies have proven that the performance of the air electrode, which is generally made up of a random physical adsorption mixture of a carbon particle material, a binder, and a catalyst, can strongly influence the discharge and charge capacities, and the cycle performance of the resulting Li–air battery [4,15,16]. A pore diameter that is too big or too small, along with large pore tortuosity results in high transfer resistance of the oxygen and lithium ions in the catalyst layers, and the production of Li_2_O_2_ can further block the narrow irregular pore channels in the air electrode [17,18]. Based on electrochemical engineering theory, the low discharge and charge capacities are mainly due to a short closing time of the narrow pore channels, especially for the exterior pores in the catalyst layers of the air electrode [19,20,21]. More importantly, the stability of electrode pore structures and the extent of the recovery of electrode pore channels are also key factors influencing Li–air battery performance. Carbon particle materials have low chemical and electrochemical stability, and can readily decompose to form unstable air electrode structures when operated at a relatively high potential, and the binders used in these materials have very low electrical conductivity and poor stability, and thus may cause the air electrode to fragment [15,22]. These phenomena increase the contact resistance between the carbon particle materials and the active catalysts, thereby lowering the discharge–charge cycle performance if irrecoverable changes to the air electrode morphology occur due to poor air electrode stability. Thus, an integrated structural air electrode is crucial for enhancing the discharge and charge capacities, and the stability and morphology recovery, of the electrode, which are necessary for improving the discharge–charge cycle performance of Li–air batteries [23,24].

To fundamentally solve the traditional problems of air electrodes that arise from the irregular and random mixture of the carbon material, binder, and catalyst, our previous studies have proposed the use of a parallel micropore channel material made of carbon nanotube (CNT) arrays, due to the outstanding mass transfer and charge transfer along the large tube axis space, as shown in Figure 1 [25]. In contrast to randomly mixed carbon materials, the ordered CNT arrays have high porosity, regular pore structure, and excellent conductivity. The application of these arrays to air electrodes also offers the advantages of strong chemical and electrochemical stability, combined with good resistance to electro-oxidation decomposition during relatively high potential operation. As a result, the transfer distances of oxygen, lithium ions, and electrons are very short in the parallel straight pore channels in the air electrode, resulting in low concentration difference overpotential and ohmic overpotential in the resulting Li–air battery. Additionally, the large tube axis space can also accommodate more Li_2_O_2_, which is highly beneficial in terms of recovering the air electrode channel morphology [26]. Unfortunately, pristine CNTs have inert surfaces, because of their completely tubular structure [27,28]. The combination of pristine CNTs and an active catalyst is unstable due to the weak physical bonding force, and the catalyst will ultimately detach from the CNTs, resulting in increased contact resistance of the pristine CNTs and active catalyst, and a higher reaction overpotential in the discharge–charge processes. Therefore, the catalytic stability of current air electrodes is insufficient, and the use of an integrated catalytic air electrode with a strong chemical bonding force could greatly enhance the discharge and charge capacities, and the cycle performance of Li–air batteries [29,30,31]. Recently, nitrogen-doped carbon nanotubes (N-CNTs) have displayed significant ORR catalytic activity in Li–air batteries. This may be due to the doping of nitrogen atoms in the graphitic layers via stable chemical bonding to form highly active pyridinic nitrogen groups that lower the reaction overpotential [32,33,34]. For instance, Li et al. [32] synthesized unordered N-CNT materials using a chemical vapor deposition (CVD) method. When the N-CNT materials were utilized as air electrodes for Li–air batteries, they exhibited a high discharge capacity of 866 mAh g^−1^, which was about 1.5 times that of pristine CNTs. The result indicates that the N-CNT materials have high electrocatalytic activities for the electrode reactions, thus improving the performance of the Li–air battery. Note that the doping of nitrogen atoms into the graphite layers of the CNTs can also provide obvious OER catalytic activity. The solid Li_2_O_2_ product can be effectively decomposed by OER during the charge process of a Li–air battery, which acts a crucial role for improving the next discharge–charge cycles of Li–air batteries [14,35]. Nevertheless, it is very difficult to build a material with a long-range regular 3D porous structure using unordered N-CNTs, and the use of a binder will cause the electrode to fragment easily; therefore, both of these strategies are unsuitable for use in Li–air batteries [36,37]. Thus, configuring an integrated structural air electrode composed of ordered N-CNT arrays, which have stable parallel micropore channels and excellent ORR/OER electrocatalytic performance, would speed the transfer of oxygen, lithium ions, and electrons, and increase the reaction activity of the resulting Li–air battery.

In the present study, we have successfully prepared a stable integrated structural air electrode made of highly catalytic N-CNT arrays on a carbon paper substrate with a conductive carbon-black layer via the catalyst seed-impregnated CVD method, as shown in Figure 2. This produces a material with an ideal parallel micropore channel structure, high electrode stability, and, unlike traditional random mixed carbon particle materials, excellent electrocatalytic activity, and sufficient storage space, to enhance the discharge and charge capacities, and the cycle performance of Li–air batteries. Additionally, the conductive carbon-black layer forms a tight connection between the N-CNT arrays and carbon paper substrate to reduce their contact resistance. It acts not only as a gas diffusion layer to make the gas distribution more homogeneous, but also as a growth substrate for the N-CNT arrays that can ensure synchronous growth and improve the distribution density of the N-CNT arrays. At the same time, the use of the carbon paper substrate as the current collector meets the requirements of enabling mass transfer to cross them [25]. In this study, the physicochemical properties of N-CNT arrays with different N/C feed ratios were extensively characterized using X-ray diffraction (XRD), Raman spectroscopy, scanning electron microscopy (SEM), transmission electron microscopy (TEM), Fourier transform infrared spectrometry (FTIR), and X-ray photoelectron spectroscopy (XPS). The electrochemical performance of Li–air batteries fabricated using the N-CNT arrays with different N/C feed ratios was elaborately investigated using galvanostatic discharge–charge tests, electrochemical impedance spectroscopy (EIS), and cyclic voltammetry (CV).

## 2. Experimental

### 2.1. Preparation of N-CNT Arrays on a Carbon-Black Layer/Carbon Paper Substrate

The carbon-black layer/carbon paper substrate impregnated with the catalyst precursor was prepared as reported previously [25]. The carbon-black layer was made up of an oxidized carbon black, a Fe-nitrate precursor, and a silica sol solution (Sigma-Aldrich, Milwaukee, WI, USA). When heated to a high temperature, the silica sol was evaporated to dryness and converted into inorganic binder in the carbon-black layer, strongly bonding the carbon-black layer and the Fe-nitrate precursor together on one side of the carbon paper substrate. Subsequently, Fe nanoparticles were formed in the carbon-black surface layer and then acted as catalyst seeds for the initial growth of the N-CNT arrays.

After elaborately exploring different nitrogen source concentration, catalyst content, reaction temperature, gas velocity, nitrogen source species, and other variables, the N-CNT arrays were synthesized on the carbon-black layer/carbon paper substrate via the catalyst seed-impregnated CVD process. Briefly, the as-prepared carbon-black layer/carbon paper substrate was placed into a quartz tube furnace (Thermconcept, Bremen, Germany) with an inner diameter of 52 mm and sintered at 350 °C for 4 h in an air atmosphere. Next, Ar gas (99.99%) was introduced into the reaction chamber at ambient pressure (flow rate: 63.3 mL min^−1^) to replace the air. The furnace temperature was adjusted to 300 °C, and then H_2_ gas (99.99%) was added to the reaction chamber for 8 h (flow rate: 160 mL min^−1^). The furnace temperature was increased to 830 °C, and the Ar and H_2_ flow rates were raised to 1500 and 210 mL min^−1^, respectively. Subsequently, the N-CNT arrays were prepared by injecting a precursor mixture consisting of 2.3 g ferrocene dissolved in 48 mL ethanediamine and xylene (Sigma-Aldrich, Milwaukee, WI, USA) into the reaction chamber at a rate of 0.8 mL min^−1^ using a syringe pump (Cole-Parmer, Chicago, IL, USA). N/C feed ratios of 1:10, 1:20, 1:30, 1:40, and 1:50 were used in the mixture precursors. After maintaining the injection rate for 60 min, growth was terminated by stopping the injection. The flow of H_2_ gas was then turned off, and the furnace was cooled to 450 °C in an Ar atmosphere. Finally, the N-CNT array/carbon-black layer/carbon paper substrate was purified by sintering at 450 °C for 2 h, and then cooled to room temperature in an air atmosphere. For comparison, a pristine CNT array/carbon-black layer/carbon paper substrate and an active carbon/carbon-black layer/carbon paper substrate were also prepared as described previously.

### 2.2. Characterizations of N-CNT Arrays

SEM images were recorded on a JSM-6700F microscope (JEOL, Akishima-shi, Japan) operated at 10 kV. These were used to investigate the morphology of the N-CNT arrays. TEM images were produced on a JEM-2010 microscope (JEOL, Akishima-shi, Japan) operated at 200 kV. These were used to show the microstructures on the interior and exterior of the N-CNTs. XRD patterns were recorded on a D/max-2500 powder diffractometer (Rigaku, Akishima-shi, Japan), operated at 40 kV and 100 mA using Cu K-α radiation sources. Raman spectra were recorded on RM-1000 Raman spectrograph (Renishaw, Gloucester, UK) with a 514.5 nm laser as excitation. These were used to investigate the crystallinity and structural properties of the N-CNT arrays. In addition, the nitrogen functional groups and contents of N-CNTs were identified by a FTIR-8400 spectrometer (Shimadzu, Kyoto, Japan) and an ESCALAB 250Xi XPS (ThermoFisher Scientific, Waltham, MA, USA) using monochromatized Al K-α radiation at 14.8 kV and 150 W.

### 2.3. Battery Assembly and Electrochemical Performance Tests

The Li–air battery consisted of a Li metal foil anode (battery grade, Merck, Munich, Germany), an organic electrolyte buffer layer [0.6 mol kg^−1^ lithium bis(trifluoromethanesulfonyl)imide (LiTFSI, Merck, Munich, Germany) in a tetraethylene glycol dimethyl ether (TEGDME, Sigma-Aldrich, Milwaukee, WI, USA) electrolyte], a Celgard 2400 lithium ion conducting membrane (battery grade, Merck, Munich, Germany), and an N-CNT array/carbon-black layer/carbon paper substrate cathode. To reduce the weight of the electrode material, the N-CNT arrays were first treated with 37% hydrochloric acid to eliminate Fe species. For comparison, batteries were also fabricated using the pristine CNT array/carbon-black layer/carbon paper substrate cathode and active carbon/carbon-black layer/carbon paper substrate cathode, respectively. A circle with an area of 1.13 cm^2^ and a diameter of 1.2 cm was cut from the N-CNT array/carbon-black layer/carbon paper substrate; this area was also defined as the effective reaction area. The Li–air battery was assembled in an Ar-filled glove box (Mbraun, Munich, Germany) with moisture and oxygen levels at less than 0.1 ppm.

Electrochemical tests of the Li–air batteries were performed in an ambient environment at room temperature (22 °C). The discharge–charge performance of the Li–air batteries was recorded using a LAND CT2001A (Wuhan LAND Electronics Co. Ltd., Wuhan, China), with a lower voltage limit of 2.0 V, and an upper limit of 4.15 V versus Li^+^/Li at a current density of 0.05 mA cm^−2^. The EIS tests of the Li–air batteries were recorded using a VMPIII Electrochemistry Workstation (Parker, Princeton, NJ, USA) in the frequency range 1 mHz to 100 kHz. The CV tests of the Li–air batteries were conducted using the same electrochemical workstation at a sweep rate of 0.1 mV s^−1^ in the voltage range 1.75–4.5 V.

## 3. Results

### 3.1. Morphology and Structure of the N-CNT Arrays

The XRD patterns of the N-CNT arrays produced using different N/C feed ratios and that of the pristine CNT arrays are shown in Figure 3A. The peaks at 2*θ* = 26.5°, 44.3°, 54.6°, and 77.9° were attributed to the graphite lattice (002), (100), (004), and (110), respectively. A sharp graphite peak with a narrow half-peak width was observed at around 2*θ* = 26.5°, which indicated the formation of multiwalled carbon nanotubes from the parallel-stacked structures of the graphite layers in N-CNT arrays [38]. In addition, the peaks at 2*θ* = 33.5°, 36.1°, and 43.2° were attributed to the Fe_3_C nanoparticle lattice (102), (020), and (121), respectively, as also shown in Figure 3A. Compared to that of the pristine CNT arrays, the N-CNT arrays had a lower peak intensity at 2*θ* = 26.5°, which confirmed the inferior crystallinity of the N-CNT arrays. Note that the peaks of the N-CNT arrays were slightly shifted to the right compared to those of the pristine CNT arrays. The higher the N/C feed ratio of the N-CNT arrays, the larger the shift of the peaks was [Figure 3A]. This result demonstrated the possible existence of reduced layer spacing in the sp^2^ graphite layers, and further corroborated the introduction of N atoms into the graphite layers of the CNTs [39].

Figure 3B shows the Raman spectra of the N-CNT arrays with different N/C feed ratios. All the N-CNT arrays had two overlapping peaks at around 1355 and 1590 cm^−1^, which are the well-known D peak and G peak related to the A_1g_ zone-edge phonon mode of a defect and the E_2g_ zone-center mode of a graphite crystal, respectively [40,41,42]. The intensity ratios of the D peak to the G peak (*I*_D_/*I*_G_) were calculated to be 0.913, 0.947, 0.906, and 0.898 for the N-CNT arrays, with N/C feed ratios of 1:10, 1:20, 1:30, and 1:50, respectively. This strongly suggested that the N-CNT arrays with N/C = 1:20 had more surface defects caused by the incorporation of a large number of nitrogen atoms into the graphite layers of the CNT arrays, whereas the other N-CNT arrays (N/C = 1:10, 1:30, and 1:50) had fewer surface defects. At an N/C feed ratio of 1:10, no additional nitrogen atoms could be incorporated into the graphite layers of the CNTs. In addition, the *I*_D_/*I*_G_ ratios of these N-CNT arrays were very close to 1, which further demonstrated the parallel-stacked structures of the graphite layers in the arrays and was consistent with our XRD results [Figure 3A].

The SEM images of the N-CNT arrays obtained with an N/C feed ratio of 1:20 are shown in Figure 4A–D. The N-CNT arrays were ordered and parallel to one another. They had a uniform length of approximately 50 μm and an average diameter of around 80 nm, as shown in Figure 4A,B. This indicated that numerous parallel straight pore channels were present in the arrays, which would allow fast mass transfer and charge transfer. Figure 4C shows the initial growth stage of the N-CNT arrays from the carbon-black layer/carbon paper substrate, suggesting that the integrated materials of the ordered N-CNT array/carbon-black layer/carbon paper substrate had excellent electrical conductivity [25]. In addition, the large inter-tube distances in N-CNT arrays are presented in the cross-sectional view in Figure 4D. The N-CNT arrays had an estimated inter-tube distance of approximately 90 nm, and an average inner-tube diameter of around 10 nm, which may significantly shorten the transfer distances of oxygen, ions, and electrons, as well as provide sufficient flexible space in the parallel straight pore channels for Li_2_O_2_ product storage.

The TEM images of the N-CNT arrays with an N/C feed ratio of 1:20 provided additional information about the N-CNT microstructures, as shown in Figure 4E,F. The tube walls had a bamboo-like structure, which is a common characteristic feature found in N-CNTs. A huge black nanocluster containing Fe, Fe_3_C, and other trace Fe species emerged at the top of the N-CNTs, as shown in Figure 4E. This indicated that the N-CNTs grew via the top growth mechanism, consistent with our previous results [25,26]. Figure 4F shows that the layer number of tube walls ranged from about 28 to 33 with a 0.36 nm inter-layer distance, and the thickness of whole tube walls is approximately 12 nm, which was largely dependent on the diameter of the N-CNTs. Additionally, a small amount of a black substance was discontinuously adhered to the surfaces of the tube walls. This was mainly ascribed to the migration of Fe_3_C nanoparticles from the top black nanocluster to the tube walls of the N-CNTs during the growth process [43].

The FTIR spectra of the N-CNT arrays obtained using different N/C feed ratios and of the pristine CNT arrays are shown in Figure 5A. Compared to those of the pristine CNTs, the spectra of the N-CNT arrays exhibited two new peaks at 1115 and 3623 cm^−1^, which are attributed to the stretching vibrations of the C–N and N–H bonds in the N-CNTs, respectively [44]. For the pristine CNT arrays, the wide and strong peak at about 1580 cm^−1^ was assigned to the stretching vibration of the C=C bond. In the N-CNT array spectra, this peak had an enhanced intensity and slightly shifted to the left with increasing N/C feed ratio. This was mainly attributed to the presence of the bending vibration of the C=N bond. The results demonstrated that the substitution of C atoms with N atoms in the sp^2^ graphitic networks induced strong infrared activity [45]. In addition, a weak peak at about 1696 cm^−1^ corresponded to the stretching vibration of the C=O bond, and a strong peak in the 2300–2400 cm^−1^ region was assigned to the asymmetric stretching vibration of the O–C=O bonds, both of which can be caused by the infrared absorption of carbon dioxide.

The XPS spectra of the N-CNT arrays with different N/C feed ratios are shown in Figure 5B,C. The wide XPS spectrum confirmed that the N-CNT arrays contained the elements C (285.6 eV), N (398.3 eV), O (531.1 eV), and Fe (711.9 eV), with surface contents of 92.48 at%, 3.18 at%, 3.45 at%, and 0.89 at%, respectively, as shown in Figure 5B. The high resolution N1s XPS spectra of the N-CNT arrays with different N/C feed ratios (1:10, 1:20, 1:30, and 1:50) show three main peaks with binding energies at around 398.3, 399.8, and 401.9 eV, which were assigned to pyridinic nitrogen groups, pyrrolic nitrogen groups, and graphitic nitrogen groups, respectively, as shown in Figure 5C [46,47]. In the range from N/C = 1:50 to N/C = 1:20, the total nitrogen content increased considerably with increasing N/C feed ratio, whereas the total nitrogen content decreased slightly from N/C = 1:20 to N/C = 1:10. The pyridinic nitrogen content followed the order (N/C)_1:20_ ≥ (N/C)_1:10_ > (N/C)_1:30_ > (N/C)_1:50_, with the maximum value at N/C = 1:20. The results indicated that the content of nitrogen functional groups in the graphitic networks increased with increasing N/C feed ratios up to 1:20, but that no further doping of nitrogen atoms into the CNTs occurred when the N/C feed ratio was increased further.

### 3.2. Electrochemical Performance of Li–Air Batteries

The performance of Li–air batteries using N-CNT arrays with different N/C feed ratios of N/C = 1:10, 1:20, 1:30, 1:40, and 1:50, pristine CNT arrays, and active carbon as the air electrode were characterized using galvanostatic discharge–charge tests at a current density of 0.05 mA cm^−2^, respectively, as shown in Figure 6. In the first cycle, the first discharge capacities were 1980, 2203, 1808, 1445, 1244, 851, and 523 mAh g^−1^, the corresponding discharge voltages were 2.79, 2.81, 2.78, 2.76, 2.75, 2.73, and 2.60 V [Figure 6A], the corresponding first charge capacities were 151, 186, 127, 103, 100, 76, and 71 mAh g^−1^, and the corresponding charge voltages were 3.68, 3.53, 3.71, 3.69, 3.75, 3.87, and 4.06 V at a uniform cut-off capacity of 60 mAh g^−1^ (Figure 6B). Obviously, the Li–air batteries with N-CNT arrays and the pristine CNT arrays exhibited higher first discharge voltages, lower first charge voltages, and larger first discharge and first charge capacities than those with active carbon. Note that the air electrode with N-CNT arrays with N/C = 1:20 had the highest first discharge voltage, lowest first charge voltage, and largest first discharge and first charge capacities compared to those of the N-CNT arrays with N/C = 1:10, 1:30, 1:40, 1:50, and pristine CNT arrays, demonstrating that the air electrode with N-CNT arrays fabricated with an N/C ratio of 1:20 could significantly enhance the first discharge–charge performance of Li–air batteries.

Figure 6C–F show the first five discharge–charge cycles and the coulombic efficiency for the first fifty cycles of Li–air batteries with the N-CNT arrays with N/C = 1:20 and the active carbon as air electrodes at a current density of 0.05 mA cm^−2^. For the Li–air battery with the N-CNT arrays, the discharge and charge capacities distinctly decreased to 106 and 28 mAh g^−1^ from the first cycle to the second, and successively decreased to 25 and 11 mAh g^−1^, respectively, over five cycles [Figure 6C]. The corresponding coulombic efficiency of the Li–air battery increased from 12% to 30% in the first and second cycles, and then exhibited a continuous but gradually slowing increasing trend to reach nearly 100% over the subsequent fifty cycles [Figure 6D]. For the active carbon, the discharge and charge capacities of the Li–air battery sharply decreased to 7 and 5 mAh g^−1^ from the first cycle to the second, respectively, and continuously decreased during subsequent cycles [Figure 6E]. The corresponding coulombic efficiency of the Li–air battery rapidly increased from 14% to 73% between the first and second cycles, and further increased to reach a plateau at around 90% over the subsequent fifty cycles [Figure 6F]. The results indicated that the discharge and charge capacities of the Li–air battery with the active carbon were extremely low in the second cycle, whereas those of the N-CNT array battery remained relatively high until the fifth cycle. Although the Li–air battery with N-CNT arrays had a higher eventual coulombic efficiency than that of the active carbon battery, both seemingly had better cycling performance in later cycles, probably because of their extremely low discharge and charge capacities. 

The EIS plots obtained for N-CNT arrays with N/C feed ratios of 1:10, 1:20, 1:30, 1:40, and 1:50, pristine CNT arrays, and active carbon as air electrodes before and after the discharge–charge cycling tests of the Li–air batteries are presented in Figure 7. The corresponding equivalent circuit is shown in the inset. The intercept of the *Z*_real_-axis at high frequency is the ohmic resistance (*R*_Ω_), which mainly includes the ionic transport resistance in the electrolyte of the air electrode pores. In the middle–low frequency regime, the semicircular loop corresponds to the charge transfer resistance (*R*_ct_) for the oxygen electro-reduction reaction at the electrode/electrolyte interface and the Warburg impedance (*Z*_w_) of the oxygen diffusion in the electrode materials, and the sloping straight line represents the diffusion capacitance (*C*_W_). The parallel constant phase element (CPE) is related to the double-layer capacitance at the surface of the electrode materials [48,49,50]. Figure 7A shows that the semicircular loops before discharge–charge cycling followed the order (N/C)_1:20_ < (N/C)_1:10_ < (N/C)_1:30_ < (N/C)_1:40_ < (N/C)_1:50_ < pristine CNTs < active carbon. The corresponding *R*_Ω_ values were 15.3, 16.3, 36.6, 61.3, 58.3, 52.2, and 61.5 Ω, the corresponding *R*_ct_ values were 39.7, 57.9, 157.1, 240.9, 320.2, 443.6, and 373.0 Ω, and the corresponding *Z*_w_ values were 27.7, 32.9, 34.5, 86.1, 60.2, 75.1, and 189.8 Ω, as shown in Table 1. Among these, the *R*_Ω_ and *Z*_w_ values for the air electrodes with the N-CNT arrays and pristine CNT arrays were smaller than those for the active carbon electrode, while the *R*_ct_ value for the air electrode with the active carbon was higher than those of the N-CNT arrays, but lower than that of the pristine CNT arrays. The results indicated that the air electrodes with the N-CNT arrays and pristine CNT arrays with parallel micropore channels had relatively lower ionic transport and oxygen diffusion resistances than those of the active carbon electrode. Additionally, the charge transfer resistances of the air electrodes with high nitrogen contents were obviously smaller than those with low nitrogen contents; that is, the oxygen reactivity increased considerably with increasing nitrogen content. Note that the oxygen reduction activity of the air electrode with active carbon was superior to that of the pristine CNT array electrode, mainly because of the larger specific surface area of the active carbon [24]. For N-CNT arrays with N/C = 1:20, the *R*_ct_ values were 239.9, 895.9, 1723.7, 2125.9, and 2483.7 Ω, and the *Z*_w_ values were 404.3, 665.1, 1651.6, 2691.3, and 2978.9 Ω, after the first, second, third, fourth, and fifth discharge–charge cycles, respectively, as shown in Table 2. This result indicated that the oxygen electro-reduction catalytic activity obviously decreased and the oxygen diffusion resistance continually increased during the discharge–charge cycling of the Li–air batteries.

The CV curves of the Li–air batteries with N-CNT arrays with N/C feed ratios of 1:10, 1:20, 1:30, 1:40, and 1:50, pristine CNT arrays, and active carbon as air electrodes before and after the discharge–charge cycling tests are shown in Figure 8. Each CV curve contained a couple of main redox peaks in the range from 1.75 to 4.5 V [16,42]. Figure 8A shows that the reduction peak currents of the N-CNT arrays with N/C = 1:10, 1:20, 1:30, 1:40, and 1:50, the pristine CNT arrays, and the active carbon were detected at −30.2, −33.9, −24.4, −23.8, −15.9, −12.3, and −14.0 mA, whereas the corresponding oxidation peak currents were observed at approximately 13.1, 19.7, 10.2, 7.0, 6.9, 3.2, and 3.3 mA, respectively. The heights (and areas) of the redox peaks followed the order (N/C)_1:20_ > (N/C)_1:10_ > (N/C)_1:30_ > (N/C)_1:40_ > (N/C)_1:50_ > active carbon > pristine CNTs at a current density of 0.1 mV s^−1^, as shown in Figure 8A. This result demonstrated that the air electrodes with N-CNT arrays had higher ORR/OER catalytic activity than the active carbon electrode, while the air electrode with active carbon had a higher ORR/OER catalytic activity than the pristine CNT array electrode, which is inversely proportional to the above *R*_ct_ value results (Table 1). In addition, the air electrode with N-CNT arrays with N/C = 1:20 displayed the highest ORR/OER catalytic activity, primarily because it contained more nitrogen groups in the graphitic layers, while the air electrode with N-CNT arrays with N/C = 1:10 showed slightly lower redox peak heights (or areas), probably because fewer nitrogen atoms were incorporated in the graphite layers of the CNTs to form nitrogen functional groups.

For the N-CNT arrays (N/C = 1:20), after the first, second, third, fourth, and fifth discharge–charge cycles, the reduction peak currents were detected at −11.4, −10.1, −8.3, −6.7, and −3.3 mA, and the oxidation peak currents were observed at approximately 7.5, 7.2, 5.9, 3.4, and 2.8 mA, respectively, as shown in Figure 8B. These redox peak heights (or areas) exhibited a downward trend after discharge–charge cycling; thus, the trend in the catalytic activity observed via CV was in accordance with the EIS results (Table 2). Note that after the fifth discharge–charge cycle, the CV curve had an approximately parallel shape with very weak redox peaks, indicating that the ORR/OER catalytic activities were extremely low, and that subsequent discharge–charge processes deteriorate the Li–air battery. More importantly, all the oxygen reduction peak heights/areas of the air electrode were dramatically higher than the oxygen evolution peak heights/areas during cycling in our experiments. This result implied that the amount of electrical charge transferred during the discharge process was much larger than that transferred during the charge process in the same cycle, primarily because the activity of the electrocatalyst in the ORR was superior to that in the OER [51]. In addition, the formation of the insoluble solid product Li_2_O_2_ in the electrolyte occurred due to inadequate decomposition because of the incomplete charge process within a cycle, which resulted in the accumulation of a large quantity of Li_2_O_2_ on the surface of the active catalyst and in the electrode pores.

## 4. Discussion

The Li–air batteries in this study had identical anodes, membrane materials, electrolytes, assembly processes, and other parameters; the only difference among the batteries was the air electrode used. Accordingly, the differences in their performance should mainly be attributed to the differences in the performance of their air electrodes [11,12]. It has been proposed that the apparent electrode potential in the discharge (charge) process of a Li–air battery is the difference (sum) of the theoretical electrode potential, concentration difference overpotential, reaction overpotential, and ohmic overpotential of the air electrode [52,53], which is described by
$$V_{cell}=E_{cell}\mp \left(\eta_{xover}+\eta_{cat}+\eta_{ohmic}\right)$$ where *V_cell_* is the apparent electrode potential at the reference current density, *E_cell_* is the theoretical electrode potential, and *η_xover_*, *η_cat_*, and *η_ohmic_* are the concentration difference overpotential, reaction overpotential, and ohmic overpotential of the air electrode, respectively. The concentration difference overpotential is substantially limited by the oxygen concentration and oxygen diffusion coefficient in the electrolyte and electrode pore structure. The reaction overpotential is fundamentally determined by the catalytic activity of the air electrode. The ohmic overpotential is affected by both the bulk electrolyte resistance and the electrode pore structure [54]. Based on electrochemical engineering theory and previous research work, shorter mass transfer and charge transfer distances in the pore channels of an air electrode lead to a lower concentration difference overpotential and ohmic overpotential for a Li–air battery given an identical electrolyte [19,25]. In this paper, the experimental EIS results showed that the *R*_Ω_ and *Z*_w_ values obtained using pristine CNT arrays with parallel straight pore channels were distinctly smaller than those obtained using active carbon with narrow irregular pore channels as the air electrode, as shown in Table 1. The experimental results are fully in accordance with the above theoretical predictions. Thus, the air electrode with parallel micropore channels and a shorter transmission distance successfully decreased the oxygen diffusion resistance and lithium ion transfer resistance, enhancing the discharge–charge performance of the Li–air batteries.

Theoretically, based on these results, assuming that the sum of the concentration difference overpotential, reaction overpotential, and ohmic overpotential is smaller, an air electrode should have a smaller apparent electrode overpotential at an identical current density; namely, the produced higher discharge voltage and lower charge voltage for the Li–air battery [52,53]. In this study, the sum of the *R*_Ω_, *R*_ct_, and *Z*_w_ values calculated from the EIS results for the N-CNT arrays with N/C = 1:10, 1:20, 1:30, 1:40, and 1:50, and the pristine CNT arrays were 107.1, 82.7, 228.2, 388.3, 438.7, and 570.9 Ω, respectively, as shown in Figure 7A and Table 1, while the corresponding discharge–charge results demonstrated that the N-CNT arrays with different N/C feed ratios had a higher discharge voltage and lower charge voltage than those of the pristine CNT arrays, as shown in Figure 6A,B. These results are consistent with the above theoretical predictions, which indicates that N-CNT arrays with higher ORR/OER catalytic activities can simultaneously lower the sum of the concentration difference overpotential, reaction overpotential, and ohmic overpotential, thus increasing the discharge voltage and decreasing the charge voltage for a Li–air battery. In addition, theoretically, for a fixed current density and uniform cut-off voltage, the higher discharge voltage and lower charge voltage would produce larger discharge and charge capacities for the same voltage falling or rising slope, respectively [25,55]. As shown in Figure 6A,B, the air electrode with the N-CNT arrays (N/C = 1:20) presented a higher discharge voltage and lower charge voltage than those of the pristine CNT arrays, while the air electrode with the pristine CNT arrays had a higher discharge voltage and lower charge voltage than those of the active carbon electrode. Thus, the air electrode with the N-CNT arrays (N/C = 1:20) had greater discharge and charge capacities compared to those of the pristine CNT arrays and active carbon. The experimental results were also consistent with the above theoretical prediction, demonstrating that the air electrode with the N-CNT arrays, which had a higher discharge voltage and lower charge voltage, could prolong the end time of the discharge and charge processes, thereby improving the discharge capacity and charge capacity of the resulting Li–air battery.

However, a large amount of the solid product Li_2_O_2_ accumulated in the electrode pore channels, due to the weak solubility of Li_2_O_2_ in the organic electrolyte [56]. Thus, the active surfaces of the air electrode became severely covered during successive discharge processes [57]. The resistance to oxygen diffusion and lithium ion transport increased sharply in the air electrode, and the ORR/OER catalytic activities continually decreased during the subsequent discharge–charge cycles, until the catalyst became inactive (Figure 7B and Figure 8B). It can be also seen from Figure 6C,E that the discharge and charge capacities of the Li–air battery with the N-CNT arrays were relatively high until the fifth cycle, whereas those of the active carbon battery were extremely low by the second cycle. The corresponding coulombic efficiency of the N-CNT arrays was eventually higher than that of active carbon, as shown in Figure 6D,F. This occurred because the OER catalytic activity of the N-CNT arrays with N/C = 1:20 was dramatically superior to that of the active carbon (Figure 8A), thereby facilitating better decomposition of the solid Li_2_O_2_ [58]. The results demonstrate that the N-CNT arrays could catalytically decompose the Li_2_O_2_ product and effectively recover the air electrode morphology during the initial discharge–charge processes, which is favorable in terms of improving the discharge–charge cycle performance of Li–air batteries. Moreover, as shown in Figure 8A, the CV curves of the Li–air batteries indicated that their catalytic activity followed the order (N/C)_1:20_ > (N/C)_1:10_ > (N/C)_1:30_ > (N/C)_1:40_ > (N/C)_1:50_ at a current density of 0.1 mV s^−1^, while the XPS spectra demonstrated that the pyridinic nitrogen content of the N-CNT arrays followed the order (N/C)_1:20_ ≥ (N/C)_1:10_ > (N/C)_1:30_ > (N/C)_1:50_ (Figure 5C). These results confirmed that the greater pyridinic nitrogen content in the N-CNT arrays could increase the ORR/OER catalytic activities of the Li–air batteries, which was also consistent with previous results [46,47]. Nevertheless, the poorly soluble product Li_2_O_2_ completely blocked the parallel micropore channels of the air electrode after five discharge–charge cycles, thereby leading to low oxygen evolution ability for the catalyst and poor discharge–charge performance for the Li–air battery [59]. The problems should be solved by improving the activity of the OER catalyst and the solubility of the Li_2_O_2_ product in the electrolyte in the future.

## 5. Conclusions

Controlled preparation of N-CNT arrays was performed on a carbon paper substrate with a conductive carbon-black layer via catalyst seed-impregnated CVD technology, and showed strong electrochemical performance in Li–air battery applications. The improvements were mainly attributed to the parallel micropore channels, strong stability, high electrocatalytic activity, and sufficient storage space of the N-CNT arrays. When the N-CNT arrays were used as an integrated structural air electrode, they allowed for the faster transfer of oxygen, ions, and electrons and exhibited high ORR/OER activities. In particular, the N-CNT arrays fabricated using an N/C feed ratio of 1:20 exhibited higher discharge voltage, lower charge voltage, and larger discharge and charge capacities than those fabricated with N/C feed ratios of 1:10, 1:30, 1:40, and 1:50, pristine CNT arrays, and active carbon. Although the N-CNT arrays, which had a high pyridinic nitrogen content, were able to catalytically decompose the product Li_2_O_2_ and effectively recover the electrode morphology during the initial discharge–charge processes of the Li–air battery, the low-solubility product Li_2_O_2_ severely blocked the air electrode channels after five discharge–charge cycles. Thus, enhancement of the electrode stability of Li–air batteries, along with improvement of the OER catalyst and modification of the electrolyte, will be important aspects of future research. The present study demonstrates an effective strategy for producing an integrated structural air electrode with stable parallel micropore channels and excellent ORR/OER catalytic activity, which may become a new way to configure air electrode materials to improve the discharge and charge capacities and cycle performance of Li–air batteries. 

## Figures and Tables

**Figure 1 nanomaterials-09-01412-f001:**
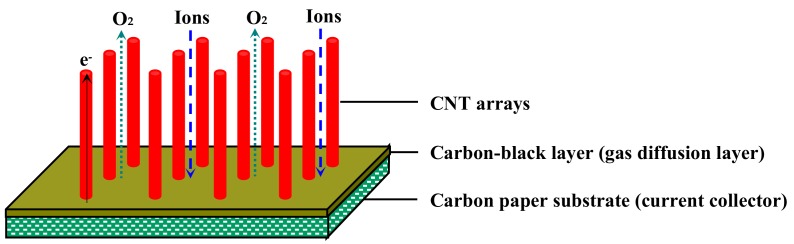
The schematic diagram of carbon nanotube (CNT) array/carbon-black layer/carbon paper substrate [25].

**Figure 2 nanomaterials-09-01412-f002:**
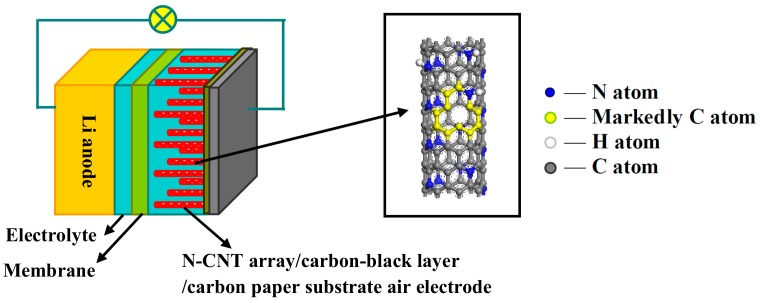
The schematic diagrams of a Li–air battery and molecular model of nitrogen-doped carbon nanotube (N-CNT) arrays containing N atoms replacing C atoms.

**Figure 3 nanomaterials-09-01412-f003:**
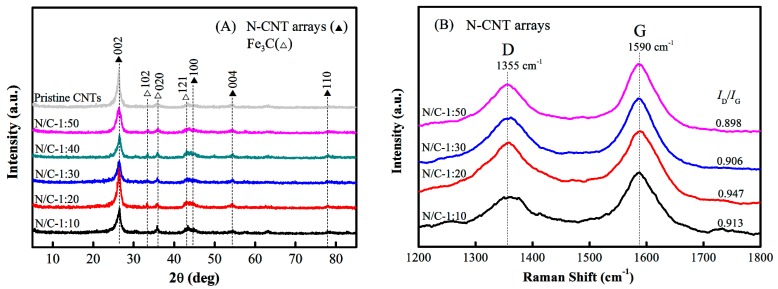
(**A**) X-ray diffraction (XRD) patterns and (**B**) Raman spectra of the N-CNT arrays and pristine CNT arrays.

**Figure 4 nanomaterials-09-01412-f004:**
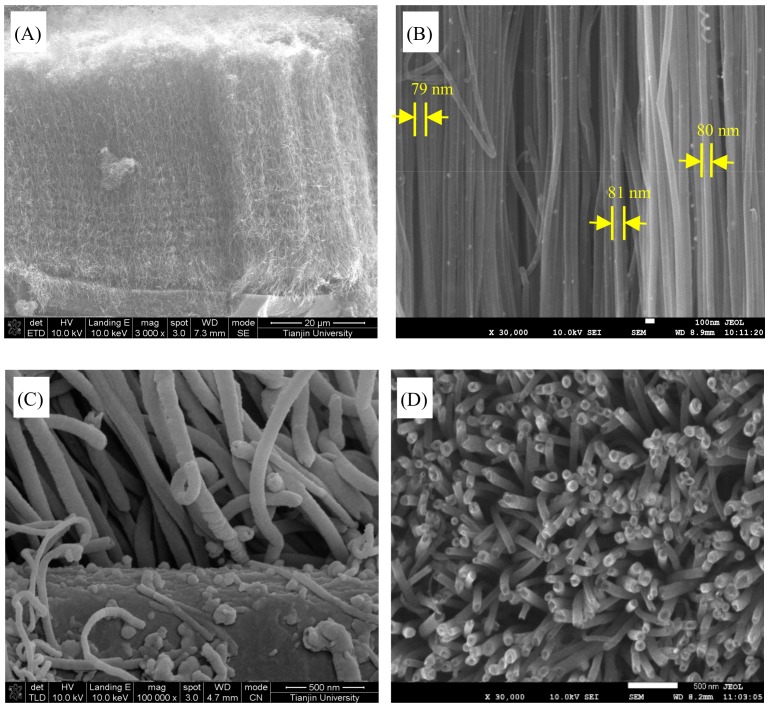
Scanning electron microscopy (SEM) images of the N-CNT arrays grown on one side surface of carbon-black layer/carbon paper substrate: (**A**) Whole micromorphology, (**B**) high magnification image, (**C**) initial stage growth, and (**D**) cross-sectional view of N-CNT array. (**E**,**F**) High magnification transmission electron microscopy (TEM) images of the N-CNTs in the arrays.

**Figure 5 nanomaterials-09-01412-f005:**
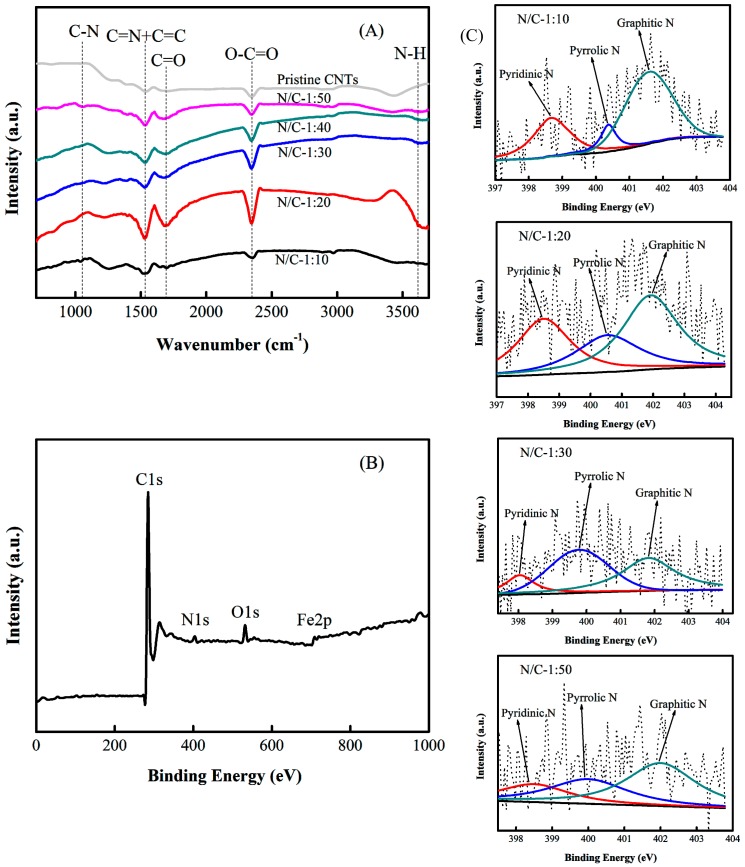
(**A**) Fourier transform infrared spectrometry (FTIR) spectra of the N-CNT arrays and pristine CNT arrays, (**B**) XPS spectra of a wide survey scan, and (**C**) high-resolution N1s XPS spectra of the N-CNT arrays.

**Figure 6 nanomaterials-09-01412-f006:**
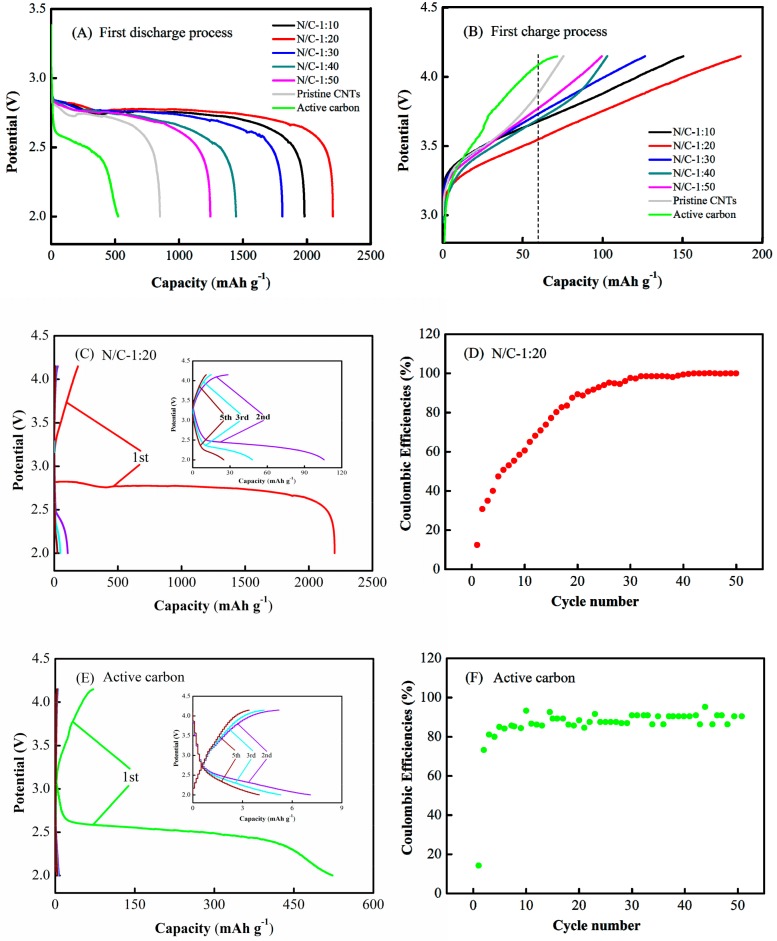
(**A**) First discharge and (**B**) first charge curves of Li–air batteries with N-CNT arrays, pristine CNT arrays, and active carbon as the air electrode, respectively. Discharge–charge curves and Coulombic efficiencies of Li–air batteries of (**C**,**D**) the N-CNT arrays with N/C feed ratio of 1:20 and (**E**,**F**) the active carbon as the air electrode at a current density of 0.05 mA cm^−2^.

**Figure 7 nanomaterials-09-01412-f007:**
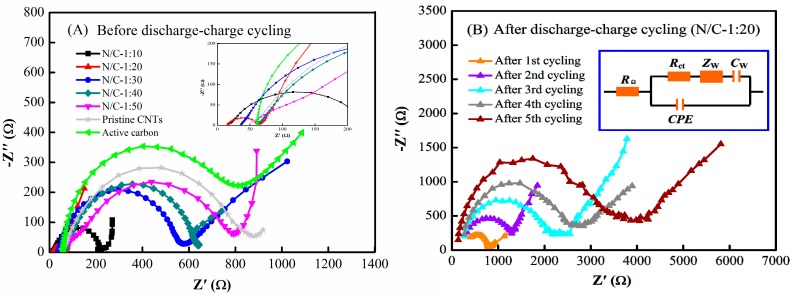
Impedance plots of Li–air batteries with (**A**) N-CNT arrays, pristine CNT arrays, and active carbon as air electrodes and the inset is the magnified impedance plots, (**B**) the N-CNT arrays with N/C feed ratio of 1:20 as the air electrode and the inset shows the corresponding equivalent circuit.

**Figure 8 nanomaterials-09-01412-f008:**
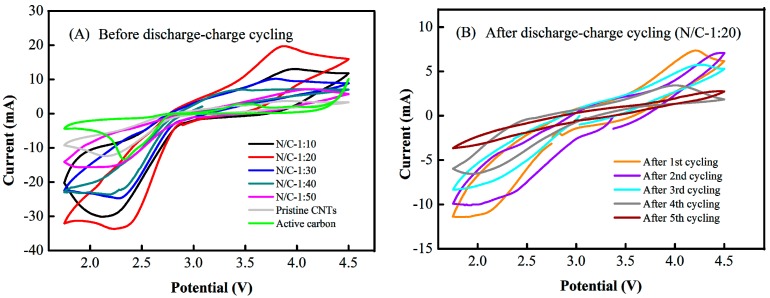
Cyclic voltammetry (CV) curves of Li–air batteries with (**A**) N-CNT arrays, pristine CNT arrays and active carbon as air electrodes, (**B**) the N-CNT arrays with N/C feed ratio of 1:20 as the air electrode at 0.1 mV s^−1^, respectively.

**Table 1 nanomaterials-09-01412-t001:** Summary of the electrochemical impedance spectroscopy (EIS) tests in the Li–air batteries before discharge–charge cycling.

Sample	*R*_Ω_ (Ω)	*R*_ct_ (Ω)	*Z*_w_ (Ω)
N-CNTs (N/C = 1:10)	16.3	57.9	32.9
N-CNTs (N/C = 1:20)	15.3	39.7	27.7
N-CNTs (N/C = 1:30)	36.6	157.1	34.5
N-CNTs (N/C = 1:40)	61.3	240.9	86.1
N-CNTs (N/C = 1:50)	58.3	320.2	60.2
Pristine CNTs	52.2	443.6	75.1
Active carbon	61.5	373.0	189.8

**Table 2 nanomaterials-09-01412-t002:** The EIS test of N-CNT arrays with N/C feed ratio of 1:20 as the air electrode after discharge–charge cycling in the Li–air battery.

Cycle	*R*_Ω_ (Ω)	*R*_ct_ (Ω)	*Z*_w_ (Ω)
After 1st cycling	248.4	239.9	404.3
After 2nd cycling	265.6	895.9	665.1
After 3rd cycling	273.9	1723.7	1651.6
After 4th cycling	276.7	2125.9	2691.3
After 5th cycling	466.7	2483.7	2978.9

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
