# Peer review of "An Integrated Structural Air Electrode Based on Parallel Porous Nitrogen-Doped Carbon Nanotube Arrays for Rechargeable Li–Air Batteries"

_nanomaterials, 2019, doi:10.3390/nano9101412_

Round 1

Reviewer 1 Report

General: Please check the grammatical errors, wherever required. English needs to be refined/polished throughout the manuscript. General: Authors are requested to unify the font usage. Experimental : What's new in this work from Ref (25)?. The preparation seems similar. Introduction: Why specifically N-doped CNTs was chosen? This needs to be clarified. Fig. 2, what’s markedly C atom? R n D: Fig. 4 (c) and (d) needs proper formatting (scale bar) Fig. 5, why there are dotted lines in deconvoluted XPS, please take care.

Reviewer 2 Report

Li et al. focussed on the preparation of CNT-arrays with integrated nitrogen doping. They characterize the arrays with suitable methods and in-depth and give a complete picture of their sample. They finally applied the N-CNT arrays as the air electrode in Li-air batteries and realized a profound performance increase.

The experiments are well designed, the manuscript is written in an easy to understanding way and the interpretation of results compared to the supplied data is sound. 

Nevertheless, I would like to recommend some improvements which may help to reach an even higher quality as compared to the current state.

I have noticed a pre-peak feature at 1250 cm-1 and a post-peakfeature at 1750 cm-1 in the Raman spectra in Fig. 3b. The latter can be attributed to a combined phonon-plasmon feature which is well excited at 514,5 nm (see Ferrari & Basco 10.1038/NNANO.2013.46) and the 1250 cm-1 (see Rebelo et al. 10.1039/C5CP06519D) related to the heteroatom doping. The question is if these feature are artifacts due to a smoothing of the spectra or a these real? In the latter case it would be worth to mention. If a smoothing function was used, please add it to the experimental section.

The main issue from my side is the quality of the XPS fits. The spectra are quite noisy which corresponds to the low quantity of of N in the CNT network. The fits made to the spectra may be partly or fully truth but because of the very small cutout, the reader will be not be able to really recognize the background feature what is an important criterion when dealing with such low intensity signals. I recommend to give a wider cutout to clearly show that the origin of the background and the fitted N-groups are suitable and fit themselves to the obtained results and that there is not doubt about it.

I would also recommend to give the concentrations of the elements C, N, O, Fe taken from the XPS and another method like CHGE and ICP-OES to show the difference between surface and "bulk" as the surface may influence the rate determining step for the oxygen reactions with both N and Fe or FeOx/FeCx.

The Nyquist plots in Fig. 7 are a bit confusing. May be it is easier to recognize the specific contributions of the samples when the authors would devide the graph in a sample with large Z' and small Z'. To my opinion this would very much help the reader to easily get the details and therewith the whole content of this figure. The same should be done with the inset of the low Ohmic region.

As the electrodes are arrays it would be easier to compare their performance to other array-like electrodes when the capacity is additionally given in mAh/cm2.

Small technical:

On page 7 second line the unit missing behind the 33, which is probably [nm] and I would write inter-layer as one word.
